# A Two-Step Fusion Method of Wi-Fi FTM for Indoor Positioning

**DOI:** 10.3390/s22093593

**Published:** 2022-05-09

**Authors:** Shenglei Xu, Yunjia Wang, Minghao Si

**Affiliations:** 1Key Laboratory of Land Environment and Disaster Monitoring, MNR, China University of Mining and Technology, Xuzhou 221116, China; cumtxsl@cumt.edu.cn; 2School of Environmental Science and Spatial Informatics, China University of Mining and Technology, Xuzhou 221116, China; hmsi@cumt.edu.cn

**Keywords:** indoor positioning, Wi-Fi FTM, single-point positioning, improved matching positioning, fusion

## Abstract

The Wi-Fi fine time measurement (FTM) protocol specified in the IEEE 802.11-2016 standard provides a new two-way ranging approach to enhance positioning capability. Similar to other wireless signals, the accuracy of the real-time range measurement of FTM is influenced by various errors. In this work, the characteristics of the ranging errors is analyzed and an abstract ranging model is introduced. From the perspective of making full use of the range measurements from FTM, this paper designs two positioning steps and proposes a fusion method to refine the performance of indoor positioning. The first step is named single-point positioning, locating the position with the real-time range measurements based on the geometric principle. The second step is named the improved matching positioning, which constructs a distance database by utilizing the existing scene information and uses the modified matching algorithm to obtain the position. In view of the different positioning accuracies and error distributions from the results of the aforementioned two steps, a fusion method using the indirect adjustment principle is proposed to adjust the positioning results, and the advantages of the matching scene information and the range measurements are served simultaneously. Finally, a number of tests are conducted to assess the performance of the proposed method. The experimental results demonstrate that the precision and stability of indoor positioning are improved by the proposed fusion method.

## 1. Introduction

Driven by trends of the Internet of things (IoT) and intelligent equipment penetration, people’s living behavior and time spent indoors have been changing. Indoor position and navigation techniques have played a vital role in retail, marketing, transportation, healthcare, emergency and rescue, and more [1]. To provide reliable and accurate position information similar to the global navigation satellite system (GNSS) in indoor environments, various techniques including wireless fidelity (Wi-Fi) [2,3,4,5], Bluetooth low energy (BLE) beacons [6,7], radio frequency identification (RFID) [8,9], ultrasonic [10], ultra-wideband (UWB) [11,12], and pseudolite [13,14] have been attempted. Solutions based on these studies behave well in certain circumstances but suffer limited accuracy and coverage.

Among these techniques, Wi-Fi-based positioning techniques are the most popular indoor positioning method due to the wide deployment of Wi-Fi access points (APs) everywhere. With RSSs from nearby APs, the distance to each AP can be calculated by utilizing the lognormal model, and subsequently, the position of the device is obtained by applying trilateration techniques [15]. This method needs a low computation, however, with the fluctuation and unpredictability of RSS, the accuracy of this method is limited. Another approach is to collect RSSs at known locations to construct the fingerprint database, and the target is located by matching the real-time RSSs with the database [16,17]. This technique is popular due to its simplicity but is time-consuming and labor-intensive. With more fine-grained channel state information (CSI) open to access, a high-precision fingerprint database can be constructed to achieve high positioning accuracy [18,19,20,21]. To be able to find a balance between accuracy and efficiency, time-based techniques including the time of arrival (TOA)/time of flight (TOF) and time of difference arrival (TDOA) are critical for ranging with high accuracy in early time [22,23,24]. For one-way time measurements of these techniques, it is difficult to realize strict time synchronization and implement it on smartphones. Then, a new protocol of the fine time measurement (FTM) specified in the 802.11-REVmc2 stand in 2016 has received increasing attention due to its capability to measure accurate ranging [25]. The protocol provides a new way to estimate the round-trip time (RTT) of the signal flight between the transmitter and the receiver without considering the clock synchronization. To apply this technique widely, the Android Pie (Android 9.0 and later ) operating system released by Google implemented this specification and the interfaces for the developers at the software level were also provided [26]. Compulab’s Wi-Fi Indoor Location Device (WILD), Google Wi-Fi routers, and Nest Wi-Fi routers support the protocol as well [27,28,29].

As a common problem of wireless signals, RTT measurement also suffers from reflection, fading, and shadowing in the indoor environment. Analyzing the composition, characteristics, and distribution of the errors is helpful to construct an accurate ranging model. Research has been conducted on the factors that affect FTM signal propagation including the clock skew, bandwidth effect, sampling effect, multipath effect, and position-dependent error in different scenarios [30,31,32,33]. However, some of these errors are hardware related and the compensation algorithms have been incorporated into the firmware. Most of the remaining errors are caused by the multipath propagation or non-line-of-sight (NLOS) effect, which is strongly correlated with the environment. Therefore, an accurate ranging model is context dependent. Guo et al. introduced an algorithm based on the Kalman filter to fuse the Wi-Fi RTT and RSS to improve the accuracy of the range measurement [34]. The polynomial fitting method was proposed in Sun et al.’s work [35]. Cao et al. proposed a range calibration model for different scenarios based on the gaussian process regression [36]. In addition, other studies concentrate on combining methods such as pedestrian dead reckoning (PDR) and map matching for improving the performance of Wi-Fi FTM positioning [35,37,38,39]. These methods locate the target based on geometric principles without making full use of the information. To compensate for this shortcoming, this paper designed two positioning steps and proposed a fusion method based on the same range measurement set to refine the performance of Wi-Fi-based indoor positioning. The first step is named single-point positioning, locating the position with the real-time range measurements based on the geometric principle. The second step is named improved matching positioning, which constructs a distance database by utilizing the existing scene information and uses the modified matching algorithm to obtain the position. Due to the error distribution of the two methods being different, a fusion method was proposed to adjust the positioning results based on the indirect adjustment principle [40,41,42]. Our major contributions are summarized as follows:We demonstrated the procedure of the Wi-Fi FTM protocol and analyzed the characteristics of several types of errors that may occur in the range measurement including clock drift, multipath propagation and NLOS effect, and random noise. An abstract ranging model based on the previous error was introduced.We analyzed and presented some geometric positioning methods to resolve the 2D position and a single-point positioning method with the advantages of resisting coarse ranging errors and numerical stability.Another approach called improved matching positioning was proposed. The method contains the offline stage and online stage. The offline stage was to construct the distance database by utilizing the extent of the location scenario and Wi-Fi APs coordinates. In the online stage, a modified matching method was utilized to process the real-time ranging fingerprint.To be able to combine the two methods from the perspective of the accuracy and error distribution, a fusion positioning method was proposed to adjust the positioning results.

The remainder of this paper is organized as follows: Section 2 presents the related works of Wi-Fi FTM in detail. Section 3 introduces the characteristics of ranging errors, the principles of the single point positioning, the improved matching positioning, and the fusion positioning methods. Section 4 describes the experiments and analyzes the results. Then, the discussion of the paper and the conclusions are presented in Section 5 and Section 6.

## 2. Related Works

Wi-Fi-based positioning techniques can be divided into two types: the fingerprint-based positioning method and the range-based method. For the first type of method, the RADAR system was the first fingerprint system, developed by Bahl et al. for indoor localization by utilizing a KNN algorithm [43]. Since then, many researchers have devoted themselves to the improvement of Wi-Fi RSS signal collection, fingerprint database construction, and positioning methods to promote the development of fingerprint indoor positioning. Song et al. analyzed the interference between Wi-Fi APs and proposed a novel AP selection method based on the relief and correlation coefficient method to select the appropriate AP signal [44]. Bi et al. introduced a fast construction method by using an adaptive path loss model interpolation to construct the fingerprint database in large-scale buildings combined with the current powerful deep learning methods [17]. Chen et al. designed a local feature-based deep long short-term memory (LF-DLSTM) approach to extract robust local features for Wi-Fi fingerprint indoor localization [45]. These studies involve each part of the fingerprint positioning technique. Moreover, fine-grained CSI measurements were used for locating the device with higher accuracy and stability. Wang et al. presented a novel deep-learning-based indoor fingerprint system based on CSI which effectively reduced location errors [19]. Wu et al. presented a fingerprint-based device-free system that enabled precise localization in indoor spaces by aggregating the CSI measurements based on Bayes classification [20]. However, the fingerprint-based methods have to construct the fingerprint database offline and update it constantly, which is time-consuming and labor-intensive, becoming a factor restricting the development of this kind of method.

For the second type of method, the Wi-Fi range-based methods mainly depended on the RSS inversion or time-related measurement. The logarithmic path-loss model is usually applied to map the relationship between RSS and ranging. These studies analyzed the features of signal strength from multiple aspects and adopted different methods to construct the logarithmic path loss model that suited the environment. After obtaining the accurate ranging, the target position was estimated by the multilateration method. This method can directly utilize RSS signals without additional equipment or protocols, but due to signal volatility, it is difficult to obtain accurate and stable ranging models. Therefore, time-related Wi-Fi positioning techniques were introduced. For example, the signal of the time of difference arrival (TDOA) and time of arrival (TOA) protocols was utilized to construct the ranging model to locate the device in early time [22,24]. Prieto et al. implemented an autonomous positioning technique based on IEEE 802.11 RTS/CTS two-frame exchange in a real scenario in 2009 [46]. Banin et al. introduced the Wi-Fi time of flight (TOF) protocol and described the development stages from the aspects of protocol conception, algorithm construction, and simulation to deployment in 2013 [23]. These related works show this technology is very attractive, even if it is difficult to implement on smartphones and requires special hardware. In 2016, a new two-way ranging approach provided by the Wi-Fi FTM protocol specified in EEE 802.11-REVmc2 attracted many scholars to participate in research. Ibrahim et al. analyzed the key factors and parameters that affected the Wi-Fi FTM ranging performance and introduced a general, repeatable, and accurate measurement framework to evaluate the time-based ranging system that was deployed at an open platform [30]. Dvorecki et al. utilized several machine learning approaches to estimate the Wi-Fi RTT ranging in contrast with the “ground-truth” information collected using a LiDAR system. The experimental results showed that the artificial neural network (ANN) approach outperformed the accuracy of the other method [47]. Choi et al. pointed out that the distance estimation strategy of range-based positioning techniques should adaptively change depending on the environment. They proposed an unsupervised learning approach based on the neural network (NN) to estimate the distance. The result illustrated that the proposed method can output the estimated distance and its standard deviation [48]. Cao et al. utilized the Gaussian process regression (GPR) to construct the position model for harsh environments and estimated the three-dimensional position in another literature by utilizing the information of the Wi-Fi FTM and estimated the 3D position of the target based on the stand particle swarm optimization algorithm based on the Wi-Fi FTM in another paper [36,49]. Si et al. made the positioning result more accurate and stable after identifying the NLOS errors [50]. The key to these methods is to construct an accurate ranging model in a particular scenario through the models.

To improve the accuracy and stability of positioning, it is more advisable to use other favorable information for fusion positioning. In view of the fusion positioning of Wi-Fi FTM, there are also many related studies. Banin et al. presented a fusion method that combined the Bayesian approach and additional map information to provide a more accurate position estimation [37]. Guo et al. introduced the clock skew, analyzed the distribution of RTT ranging error, and presented a calibration method to eliminate the RTT range offset. An integrated ranging algorithm based on the Kalman filter to fuse the Wi-Fi RTT and RSS inversion ranging was proposed to enhance the scalability and robustness of the positioning system [34]. Sun et al. proposed a new tightly coupled positioning algorithm based on an extended Kalman filter to fuse the Wi-Fi FTM and PDR data to improve accuracy [35]. Choi et al. proposed a calibration-free positioning technique using Wi-Fi ranging and PDR, where every parameter in the system is optimized in real time. The experimental results demonstrated that the proposed method achieved good performance [39]. Huang et al. presented the idea of the ranging fingerprint in their paper and built a fingerprint database through a CNN method. A particle filter was utilized to integrate the Wi-Fi ranging fingerprint, PDR, and map information to achieve sub-meter-level positioning. The results showed that the integrated system could obtain the mean positioning error of 0.41 m [38]. For the positioning accuracy of different methods, Gustafsson et al. presented models for RSS, TOA, and angle of arrival (AOA) measurements and proposed the position accuracy boundary criterion accordingly based on the Cramer–Rao lower bounds (CRLB) [51]. Patwari et al. focused on the shorter-range, low-antenna, sensor network environment to construct the CRLB model to help system designers and researchers select measurement technologies and evaluate localization algorithms [52].

In summary, these studies focused on ranging error analysis and compensation, ranging model construction under certain circumstances, and positioning methods integrating other signals for indoor positioning. Compared with other positioning techniques, Wi-Fi FTM is worth studying as a Wi-Fi-based positioning technique due to its convenience.

## 3. Materials and Methods

### 3.1. Ranging Model Based on Wi-Fi Fine Time Measurement

The Wi-Fi FTM protocol enables a new two-way ranging approach between initiator and responder. The initiator is usually a smartphone (SP), and the responder is a Wi-Fi access point (AP), both of which support the FTM protocol. The procedure of the FTM protocol is shown in Figure 1.

The whole procedure starts with the SP sending an FTM request to the AP and waiting for the acknowledgment (Ack) packet from the AP. Then, one measurement of the two-way ranging occurs as follows: the AP sends the FTM frame to SP and records the time of departure (TOD) $\;t_{1}$. The SP receives the FTM frame and records the time of arrival (TOA) $\;t_{2}$. Then, it sends the Ack instruction back to the AP after processing the frame and records TOD $\;t_{3}$. The AP receives the Ack information and records the TOA $\;t_{4}$. During the next measurement, the times $\;t_{1}$ and $t_{4}\;$are sent to the SP. The ideal time of flight between two devices is denoted as $\;t_{f}$ which can be extracted as follows:
(1)$$\;t_{f}=\frac{\left(\;t_{4}-t_{1}\right)-\left(\;t_{3}-t_{2}\right)}{2}=\frac{\;t_{ap}-t_{sp}}{2}\;$$
where $t_{sp}$ and $t_{ap}$ represent the time delay of the measurement in both devices, respectively. The distance between SP and AP is estimated in Equation (2) by multiplying the time of flight $\;t_{f}$ and speed of the light c:(2)$$d=c\cdot \;t_{f}$$

The procedure demonstrates that the two-way ranging of the protocol allows two devices without common clock synchronization.

However, the independent and unsynchronized clocks of two devices are affected by imperfections in the time references. The clock drift is assumed as the dominant component in the time imperfections [53] and is modeled as the following expressions:(3)$$\;{\hat{t}}_{sp}=\left(1+e_{sp}\right)t_{sp}\;\;{\hat{t}}_{ap}=\left(1+e_{ap}\right)t_{ap}$$
where $\;{\hat{t}}_{sp}$ and $\;{\hat{t}}_{ap}$ represent the nominal time difference values in SP and the AP, respectively, and $e_{sp}$ and $e_{ap}$ model the deviation from the nominal frequency of the two devices. Based on Equations (2) and (3), the estimated time of flight $\;{\hat{t}}_{f}$ becomes
(4)$$\;{\hat{t}}_{f}=\frac{1}{2}\left(\;{\hat{t}}_{ap}-{\hat{t}}_{sp}\right)\;$$
and the error equals:(5)$$\;{\hat{t}}_{f}-t_{f}=\frac{1}{2}\left(e_{ap}t_{ap}-e_{sp}t_{sp}\right)=e_{ap}t_{f}+\frac{t_{sp}}{2}\left(e_{ap}-e_{sp}\right)\;$$

The time delay $t_{sp}$ is determined by the processing time of the packets to be received and transmitted. For the same device processing the same signal, the time delay $t_{sp}$ is assumed as a constant value and is expressed as a certain multiple of the time of flight. The error can be simplified to a variable only related to the time of flight in Equation (6):(6)$$\;{\hat{t}}_{f}-t_{f}=e_{ap}t_{f}+\frac{t_{sp}}{2}\left(e_{ap}-e_{sp}\right)=e_{ap}\;t_{f}+\frac{k\;t_{f}}{2}\left(e_{ap}-e_{sp}\right)=e_{f}t_{f}\;$$
where $e_{f}$ is the error factor including the deviation from the nominal frequency and the time delay of processing the signal. Therefore, the time of flight distance error $d_{e_{f}}$ is increased with the distance measurement.
(7)$$d_{e_{f}}=c\cdot \;e_{f}t_{f}\;$$

To verify the performance of the error $d_{e_{f}}$, a group of experiments under the condition of line-of-sight (LOS) outdoors, in which it is assumed the error of the ranging is only related to the error $e_{f}$, was carried out. Experimental results show that the performance of the distance error is consistent with the previous analysis, increased with the distance, and has a negative term. Moreover, similar to other wireless signals, FTM measurement suffers from reflection, fading, and shadowing. The multipath propagation and non-line-of-sight effects, which are caused by the surrounding environment, such as building structure, materials, and people, have significant influence on the time of signal flight. A schematic diagram of this type of error is shown in Figure 2. In the figure, the green line represents the true path and the red line represents the possible path of signal propagation. When the signal is received by the smartphone under this circumstance, the signal propagates directly through the material or spreads through the reflective surface to the smartphone. Both cases will result in longer signal flight time. Increasing the channel bandwidth or applying super-resolution algorithms can improve the device’s ability to distinguish the first arrival through a direct path to suppress the multipath propagation effect [30,54]. In addition, the RSS can be collected in the ranging procedure to help identify the NLOS effect [50].

Except for the above-introduced errors, the random error $e_{r}$, which conforms to the normal distribution, is inevitable. Therefore, taking all these error items into account, the ranging model is expressed in Equation (8):(8)$$\hat{d}=d+d_{e_{f}}+d_{e_{m}}+d_{e_{r}}$$
where $\;\hat{d}$ is the observed distance, d is the actual distance, and $d_{e_{f}}$, $\;d_{e_{m}}$, and $d_{e_{r}}$ are distance errors caused by $e_{f}$, $e_{m}$, and $e_{r}$. Among these errors, the errors $e_{f}$ and $e_{m}$ have rules to follow and can be suppressed easily. Due to the complexity of the indoor environment, the error $e_{m}\;$ is complex and varies with different scenarios, which is difficult to eliminate.

### 3.2. Single Point Positioning Method

After getting the range measurements, the position can be resolved by geometric positioning. To achieve a bi-dimensional (2D) position, at least three Wi-Fi FTM range measurements from different APs are required. Figure 3 shows the principle of the geometric positioning method in a 2D plane.

Suppose the coordinate of the SP is S=(x,y) and the distance to $i_{th}$ AP, whose position is $\left(x_{i},y_{i}\right)$, is $d_{i}$. The distances from SP to each AP can be calculated by the Euclidean distance in the following form:(9)$$\{\begin{array}{c} \sqrt{{\left(x_{1}-x\right)}^{2}+{\left(y_{1}-y\right)}^{2}}=d_{1}\\ \cdots \\ \sqrt{{\left(x_{n}-x\right)}^{2}+{\left(y_{n}-y\right)}^{2}}=d_{n} \end{array}$$

Due to the errors in the range measurements, Equation (9) is squared and simplified to a linear equation and is written as follows:(10)$$A_{\left(n-1\right)\times 2}\times S-L_{\left(n-1\right)}=V_{\left(n-1\right)}\;A_{\left(n-1\right)\times 2}=2\left[\begin{array}{cc} \begin{array}{c} x_{1}-x_{2}\\ \vdots \\ x_{1}-x_{n} \end{array} & \begin{array}{c} y_{1}-y_{2}\\ \vdots \\ y_{1}-y_{n} \end{array} \end{array}\right]\;L_{\left(n-1\right)}=\left[\begin{array}{c} d_{2}^{2}-d_{1}^{2}-\left(x_{2}^{2}-x_{1}^{2}\right)-\left(y_{2}^{2}-y_{1}^{2}\right)\\ \vdots \\ d_{n}^{2}-d_{1}^{2}-\left(x_{n}^{2}-x_{1}^{2}\right)-\left(y_{n}^{2}-y_{1}^{2}\right) \end{array}\right]$$
where V is the distance measurement noise. The object of the Equation (10) is to minimize the value $V^{T}V$. The S is commonly resolved by using the least square (LS) method, which is expressed in Equation (11):(11)$$S={\left(A^{T}A\right)}^{-1}A^{T}L$$

This approach is easy to implement and run fast, which is called the multilateration positioning method. It is the most commonly used method in the distance-based positioning system.

Another way is to perform the Taylor series of nonlinear Equation (9) at the approximate coordinate $S_{0}=\left(x_{0},y_{0}\right)$ and resolve the result iteratively by the Newton method. The details are described as follows. Firstly, Equation (9) is converted to Equation (12) in the form of first-order Taylor series:(12)$$\{\begin{array}{c} \frac{x-x_{0}}{d_{0}^{1}}dx+\frac{y-y_{0}}{d_{0}^{1}}dy=d_{1}-d_{0}^{1}\\ \cdots \\ \frac{x-x_{0}}{d_{n}^{1}}dx+\frac{y-y_{0}}{d_{n}^{1}}dy=d_{n}-d_{0}^{n} \end{array}\;d_{0}^{i}=\sqrt{{\left(x_{0}-x_{i}\right)}^{2}+{\left(y_{0}-y_{i}\right)}^{2}}$$
where $d_{0}^{i}$ is the distance from the approximate position $S_{0}$ to the $i_{th}$ AP and ∆=[dx,dy] are the corrections coordinates of $S_{0}$. Similarly, Equation (12) can be written in matrix form because of the random range measurement errors:(13)$$V_{n\times 1}=A_{n\times 2}\times {∆}_{n\times 1}-L_{n\times 1}\;A_{n\times 2}=\left[\begin{array}{cc} \begin{array}{c} \frac{x_{1}-x_{0}}{d_{0}^{1}}\\ \vdots \\ \frac{x_{n}-x_{0}}{d_{n}^{1}} \end{array} & \begin{array}{c} \frac{y_{1}-y_{0}}{d_{0}^{1}}\\ \vdots \\ \frac{x_{n}-y_{0}}{d_{n}^{1}} \end{array} \end{array}\right]\;L_{n\times 1}=\left[\begin{array}{c} d_{1}-d_{0}^{1}\\ \vdots \\ d_{n}-d_{0}^{n} \end{array}\right]$$

The corrections ∆ can also be solved by the least square (LS) method, as shown in Equation (14) and estimation of S is obtained by Equation (15):(14)$$∆={\left(A^{T}A\right)}^{-1}A^{T}L$$
(15)$$S=S_{0}+∆$$

The procedure needs to be iterated multiple times and will not stop until the corrections ∆ fall below a threshold or the iteration number is larger than a certain value. This method is called the single-point positioning method and the pseudocode of this method is shown in Table 1.

Compared with the multilateration positioning, the single-point positioning can resist coarse ranging errors and has the characteristic of numerical stability. The single-point positioning method is a better choice for geometric positioning in a complex indoor environment.

### 3.3. Improved Matching Positioning Method

Literature shows that the range measurements provided by Wi-Fi FTM may have standard deviations of 1–2 m under favorable circumstances [54]. Positioning based on geometric principles is still the mainstream solution of this technique. However, due to the complexity of the indoor environment, this technique needs to construct an accurate ranging model for different scenarios. In reference [36], three range compensation models were proposed for one scene. To solve this problem, this paper proposed an improved matching positioning method without needing to construct the ranging model in advance.

The proposed method is similar to fingerprint positioning containing two stages: the offline stage and the online stage. The main task of the offline stage is to construct a distance database while the online stage is real-time matching positioning. During the offline stage, the distance database is constructed based on the coordinates of APs and the extent of the location scenario. Figure 4 shows the process of constructing the distance database.

The whole process is described as follows:
Step 1: determining the extent of the current location scenario from the map. The point $\left(x_{min},y_{min}\right)$ refers to the minimum coordinate of the map whereas the point $\left(x_{max},y_{max}\right)$ refers to the maximum coordinate.Step 2: to facilitate the construction of the distance database, we choose the form of grid topology. With the minimum coordinate as the lower-left and the maximum coordinate as the upper-right to form an outer rectangle. The rectangle is then divided into grids with a certain spacing s, where it is divided into m parts along the x-axis and n parts along the y-axis. $x_{i}$ refers to the $i_{th}$ point along the x-axis while $y_{j}$ refers to the $j_{th}$ point along the y-axis. Therefore, the coordinates of the whole grid can be constructed in the matrix form (16).
(16)$$x_{i}=x_{min}+i\cdot s\;\;\;\;\;\;\;\;\;\;\;\;\;\;i\in 0,1,\ldots ,m\;y_{j}=y_{min}+j\cdot s\;\;\;\;\;\;\;\;\;\;\;\;\;\;j\in 0,1,\ldots ,n\;\left[\begin{array}{ccc} \left(x_{0},y_{n}\right) & \cdots & \left(x_{m},y_{n}\right)\\ \vdots & \ddots & \vdots \\ \left(x_{0},y_{0}\right) & \cdots & \left(x_{m},y_{0}\right) \end{array}\right]$$Step 3: suppose there are k APs in the location scenario and the coordinates of the $l_{th}\;$AP are expressed $\left(x_{l},y_{l}\right)$. The Euclidean distance $dis_{q,AP\left(l\right)}$ is calculated from $q_{th}\;$grid point $\left(x_{i},y_{j}\right)$ to $l_{th}\;$AP. The distances of $q_{th}\;$grid point to k APs form a fingerprint is expressed as (17).
(17)$$fp_{q}=\left[dis_{q,AP\left(1\right)},dis_{q,AP\left(2\right)},\ldots ,dis_{q,AP\left(k\right)}\right]$$Step 4: the whole distance fingerprints of all grid points are stored to form a distance database. The form of the distance database is illustrated in Table 2.


During the online stage, the smartphone collects real-time range measurements to match the database to resolve the position. K-nearest neighbor (KNN) is one of the most common and effective classifier algorithms.

To obtain good results, the range measurements need to be transformed during the matching process. Assuming that, in an indoor environment with a good electromagnetic environment, the error caused by multipath propagation and NLOS effects is small, the rule of transformation mainly depends on the error $e_{f}$. The range measurement $\hat{d}$ is considered to be composed of the true distance d and the error $d_{\;e_{f}}$, which can be expressed as Equation (18):(18)$$\hat{d}\approx d+d_{\;e_{f}}+d_{\;e_{r}}$$

According to the previous error analysis, the range measurement $\hat{d}$ requires to be normalized as $Norm_{\;e_{t}}$ with the expected value in Equation (19) to remove the influence related to the distance:(19)$$E(Norm_{\;e_{t}})=E\left(\frac{\hat{d}}{d}-1\right)=E\left(\frac{d+d_{\;e_{f}}+d_{\;e_{r}}}{d}\right)-1=E\left(\frac{c\cdot \;e_{f}\;t_{f}}{c\cdot \;t_{f}}\right)=e_{f}$$

When we stand at the $q_{th}$ grid point to survey the distance to the $l_{th}$ AP, the true distance is $dis_{q,AP\left(l\right)}$ and the real-time distance is $dis_{r,AP\left(l\right)}$. Therefore, in the process of real-time positioning, when the user moves to a position, the sum of errors with the real-time fingerprint matching with some items in the database is the minimum:(20)$$error=\overset{k}{\underset{i=1}{\sum }}\sqrt{{\left(\frac{dis_{r,AP\left(i\right)}}{dis_{q,AP\left(i\right)}}-1\right)}^{2}}=\overset{k}{\underset{i=1}{\sum }}\sqrt{{\left(\;e_{f}\right)}^{2}}=\overset{k}{\underset{i=1}{\sum }}\left|\;e_{f}\right|\;q\in 0,1,\ldots ,m\times n$$

The real-time ranging fingerprint matches all items in the database and we choose the first K items of the small sum errors averaging the coordinates to obtain the final positioning result which is shown in Equation (21):(21)$$\left(x,y\right)=\frac{1}{K}\overset{K}{\underset{i=1}{\sum }}(x_{i},y_{i})$$

Compared with the single-point positioning method, the principle of the improved matching positioning is more concise and has the extra constraint of the spatial grid, which has a certain constraint effect on coarse errors. However, this method requires the construction of spatial grids in advance, and the positioning performance will be different with the resolution of grids changing.

### 3.4. Fusion Positioning Method

In this paper, two indoor positioning methods based on Wi-Fi FTM range measurements are introduced, respectively. Because the principles of the two positioning methods are completely different, this paper proposed a fusion positioning algorithm based on the principle of indirect adjustment to integrate the two positioning results. Before fusion, the positioning accuracy of each method should be determined in advance, which is closely related to the error variance.

Suppose that the resulting coordinate of the single-point positioning method is ${\hat{X}}_{1}$, and the distance between this position and each AP constitutes a group of pseudo-distances $L_{1}$ with the error $\varepsilon_{l1}$ and variance $\sigma_{l1}$. Then, the error $\varepsilon_{p1}$ of the positioning result is amplified by the error $\varepsilon_{l}$ according to Equation (14)
(22)$$\varepsilon_{p1}={\left(A_{1}^{T}A_{1}\right)}^{-1}A_{1}^{T}\varepsilon_{l1}$$
and the corresponding variance of $\sigma_{p1}$ can be amplified by Equation (23):(23)$$\sigma_{p1}=E\varepsilon_{p1}\varepsilon_{p1}{}^{T}=E{\left(A_{1}^{T}A_{1}\right)}^{-1}A_{1}^{T}\varepsilon_{l1}{\left({\left(A_{1}^{T}A_{1}\right)}^{-1}A_{1}^{T}\varepsilon_{l1}\right)}^{T}={\left(A_{1}^{T}A_{1}\right)}^{-1}\sigma_{l1}$$

Based on the same rules, the resulting coordinate of the improved matching positioning method is ${\hat{X}}_{2}$, the corresponding variances of pseudo distances and the positioning result are $\sigma_{l1}$ and $\sigma_{p2}$, respectively. The variance $\sigma_{p2}$ is amplified by the variance $\sigma_{l1}$ based on Equation (24):
(24)$$\sigma_{p2}{\left(A_{2}^{T}A_{2}\right)}^{-1}\sigma_{l2}$$

Since the positioning results of the two methods are obtained based on the same set of range measurements, it can be considered that the corresponding pseudo-distances accuracy is the same, that is, $\sigma_{l1}$ equals $\sigma_{l2}$ and both of them are equal to the variance of unit weight $\sigma_{0}$. Therefore, the variances in the positioning results ${\hat{X}}_{1}$ and ${\hat{X}}_{2}$ are $\sigma_{1}$ and $\sigma_{2}$, respectively, which can be expressed in Equation (25):(25)$$\sigma_{1}=trace{\left((A_{1}^{T}A_{1}\right)}^{-1})\sigma_{0}\;\sigma_{2}=trace{\left((A_{2}^{T}A_{2}\right)}^{-1})\sigma_{0}$$

The corresponding weights are set to $p_{1}$ and $p_{2}$, which can be determined by the reciprocal of the variance in the following Equation (26):(26)$$p_{1}=\frac{1}{\sigma_{1}}\;p_{2}=\frac{1}{\sigma_{2}}$$

According to the principle of the indirect adjustment, the error equations of the fusion positioning result X with ${\hat{X}}_{1}$ and ${\hat{X}}_{2}$ can be expressed in Equation (27):(27)$$\{\begin{array}{c} V_{1}=X-{\hat{X}}_{1}\\ V_{2}=X-{\hat{X}}_{2} \end{array}$$
where $V_{1}$ and $V_{2}$ are the random errors of positioning results. The indirect adjustment equation of two-precision observations is expressed in Equation (28):(28)V=AX−L
where A is a matrix composed of $A_{1}$ and $A_{2}$, in the form of ${\left[\begin{array}{cc} 1 & 1 \end{array}\right]}^{T}$. L is a matrix composed of ${\hat{X}}_{1}$ and ${\hat{X}}_{2}$, in the form of ${\left[\begin{array}{cc} {\hat{X}}_{1} & {\hat{X}}_{2} \end{array}\right]}^{T}$. P is a matrix composed of $p_{1}$ and $p_{2}$, in the form of $\left[\begin{array}{cc} p_{1}\; & 0\;\\ 0 & p_{2}\; \end{array}\right]$. The objective function of Equation (29) is to minimize the error $VV^{T}$. Then, the weighted least square (WLS) principle is used to estimate the result in Equation (29).
(29)$$\hat{X}={\left(A^{T}PA\right)}^{-1}A^{T}PL$$

The fusion positioning result is finally obtained by Equation (30) after simplification:(30)$$\hat{X}=\frac{p_{1}}{p_{1}+p_{2}}{\hat{X}}_{1}+\frac{p_{2}}{p_{1}+p_{2}}{\hat{X}}_{2}$$

The variance of the fusion positioning result is calculated in (31), which is smaller than the variances of the two methods.
(31)$$\sigma_{x}=\frac{\sigma_{1}\sigma_{2}}{\sigma_{1}+\sigma_{2}}\;\sigma_{x}=\frac{\sigma_{1}}{\sigma_{1}/\sigma_{2}+1}<\sigma_{1}\;\sigma_{x}=\frac{\sigma_{2}}{1+\sigma_{2}/\sigma_{1}}<\sigma_{2}$$

The above Equation (31) shows that the fusion positioning result is more stable than the single positioning result.

Figure 5 demonstrates the process of fusion positioning.

The process consists of two stages: one is the parameter calculation and the other is the fusion positioning. During the parameter calculation stage, the improved matching positioning result and the Wi-Fi APs coordinates were utilized to calculate the coefficient matrix $A_{1}$ at the positioning result ${\hat{X}}_{1}$. The coefficient matrix $A_{2}$ was calculated in the same way. Then, both variances of the two positioning results are obtained by Equation (25) based on the coefficient matrix and the corresponding weights are the inverse of the variances. Finally, the fusion positioning result Equation (30) is derived from the principle of the indirect adjustment and obtained by a weighted average of the two positioning results.

In summary, the architecture of the two-step fusion positioning is illustrated in Figure 6. Firstly, the Wi-Fi APs coordinates and map information are utilized to construct the true distance database. Secondly, the range measurements of FTM are applied to obtain the positioning results by using the two proposed positioning approaches, respectively. Lastly, the final positioning results are resolved based on the principle of the indirect adjustment by combing the aforementioned two positioning results, making full use of the range measurements.

## 4. Experiments and Analysis

### 4.1. Environment Description and Experimental Setup

The experiments were set up on the second floor of the test site in the 54th Research Institute of China Electronics Technology Group Corporation. Figure 7 shows the experimental environment, including two rooms and a corridor adjacent to each other. Each room is 9.5 m long and 5.83 m wide, while the corridor is 19.51 m long and 1.74 m wide. The site consists of various materials, such as concrete, wood, glass, and some furniture. The Wi-Fi APs used in the experiment are equipped with the Intel Dual Band Wireless-AC 8260 to provide FTM functionality running on 2.4 GHz of default settings, and the smartphone is a Google Pixel 3 running on Android 9, which can gather the range measurement.

Three experiments were carried out at this test site. The first experiment was to analyze the features of several groups of real-time range measurements in different scenarios. The second experiment was to assess the accuracy and efficiency of the improved matching positioning method under different grid resolutions. The third experiment was to evaluate the performance of the proposed methods including the improved matching positioning, the single-point positioning, and the fusion positioning methods. To prove the superiority of the proposed methods, the other two methods multilateral positioning and WLS positioning [35], which were applied in the literature, were also conducted.

### 4.2. Features of Real-Time Range Measurements in Different Scenarios

In this experiment, three groups of real-time range measurements were collected in three scenarios including outdoor, corridor, and in the room. The AP and SP were put on tripods, 1.3 m tall, making them in the same plane. Each group of range measurements is 36 sets of data collected with the same device at 0.5 m intervals between 0.5 m and 18 m under LOS conditions. The sampling time of each data set was 5 s and the sampling frequency was 5 Hz. The range measurements were averaged per second as real-time data. Therefore, each point had five real-time range measurements in the same scenario. Figure 8 shows the observed distance, the actual distance, and the estimated distance in three scenarios.

In Figure 8, blue points and red points denote the observed distance and the true distance, respectively. The estimated result of the observed distance is represented by the green line. As can be seen from Figure 8b,c, even under LOS conditions, the distance measurements are easily affected by the multipath propagation effect, resulting in larger errors and measurement fluctuations, which is particularly obvious in the room environment.

Moreover, the distance linear models of three scenarios are in Equation (32), respectively.
(32)$${\hat{d}}_{o}=0.865\times d-0.157\;{\hat{d}}_{c}=0.870\times d-0.212\;{\hat{d}}_{r}=0.791\times d+0.458$$
where ${\hat{d}}_{o}$, ${\hat{d}}_{c}$, and ${\hat{d}}_{r}$ are the estimated distance of the outdoor setting, the corridor, and the room, respectively.

From Equation (32), the distance model is different in different scenarios. The model will be more complex and changeable especially in the indoor environment to describe the characteristics of the real-time range measurements. For positioning methods based on the geometric principle, the distance model needs to be constructed to obtain higher precision positioning results, which is a great challenge.

### 4.3. Evaluation of Improved Matching Positioning with Different Spacing

This experiment was carried out to verify the positioning performance of the improved matching positioning method under different grid spacing. There are 4 Wi-Fi APs with the same configuration deployed on tripods 1.3 m tall, making APs and SP on the same plane, to resolve the 2D position. The SP was used to collect the real-time range measurements in 1 s and the mean value of the sampling data was utilized in the algorithm.

Figure 9 shows the deployment and the test points at the test site. The specific positions of APs are marked with the red five-pointed star. Subsequently, 193 test points are designed in advance, and marked with blue squares, with an interval of 1.2 m.

To make the distance database cover the entire test site, we artificially expanded the extent of the current location scenario by 2 m upwards, east, west, south, and north. The lower-left corner of the site was made as the minimum coordinate, and the upper right corner as the maximum coordinate, with a certain spacing to construct the distance database. The certain spacing was designed as 0.2 m, 0.5 m, 1 m, 1.5 m, and 2 m. Figure 10 shows a schematic diagram of the coverage of grid points.

The distance database is automatically constructed by the program, and the range measurements of 193 test points were used for evaluating the accuracy and efficiency of the improved matching positioning method with different spacings. Table 3 shows the entry number of the distance database with different spacing and the method running time, which is the average time of 10 runs of the method. From Table 3, we can conclude that the running time is approximately proportional to the entry number.

We calculated the positioning results through the improved KNN matching method and compared them with the true coordinates of the test points to estimate the positioning error. The value of K of the KNN algorithm is 5. Figure 11 intuitively shows the performance of the improved matching positioning with different spacing by utilizing the box plot and cumulative errors plot. From the figure, the blue dotted line represents the method with a spacing of 0.2 m, red corresponds to 0.5 m, green refers to 1 m, magenta denotes 1.5 m, and black represents 2 m. It can be seen from Figure 11a that each type of positioning has some outlier errors. The position accuracy decreased with the increase in spacing and the result of 1.5 and below is close and better than that of the distance database with a resolution of 2.0 m. The blue, red, green, and magenta curves trend in Figure 11b is similar, and has a better performance than the black lines. The conclusions are consistent with Figure 11a.

Table 4 shows a more detailed positioning error comparison with data. Based on the error cumulative distribution function shown in Figure 12b, the positioning error in the top 50%, 70%, and 80% under different spacings are respectively counted, and the corresponding mean error (ME) and root mean square error (RMSE) are calculated. From the perspective of data analysis, the improved matching positioning method with a resolution of 1.5 m and below has almost a close performance in the top 50%.

From the overall ME and RMSE, the improved matching positioning method based on the distance database has the advantages of high accuracy and good stability. However, as the indoor scenario area increases, the number of grid points will grow in a multiple of squares, making calculations more time consuming. Considering the balance of operating efficiency and positioning accuracy, the improved matching positioning method with 1.5 m spacing is used in our experiments.

### 4.4. Fusion Positioning Experiment

The experimental settings were the same as described in Section 4.3. To verify the effectiveness of the proposed methods, multilateral positioning and WLS positioning were carried out on the other two methods to compare with the proposed methods based on the same real-time range measurements group. The group data were respectively composed of two sets, and each set is the mean value of the sampling data in 1 s at each test point. Among them, the weight matrix of the WLS method construction method is consistent with that adopted in reference [35], which is based on the inverse distance weights. A line chart in Figure 12 is used to show the positioning error of different methods intuitively. In Figure 12, the blue line, cyan line, magenta line, green line, and red line refer to the positioning errors of the multilateration, improved matching, single-point, and fusion positioning methods. The positioning errors of different positioning methods based on the two groups of data are similar. The blue line and cyan line have more ups and downs; the red line has fewer ups and downs and the fluctuations are more stable. The magenta line and the green line are between the two blue and red lines, and the positioning error is smaller than the blue line.

Figure 13 shows the performance of the three proposed methods from the perspective of the spatial distribution of positioning errors, which plots the positioning error on a 2D plane, and the color depth is used to represent the error size. In the figure, blue represents the smaller error, yellow represents the larger error, black points refer to the test point, and the points with the position error of 1 m are connected with contour lines. From the figure, we can see that the positioning accuracies and error distributions of the proposed methods are different. Compared with the single-point positioning and the improved matching methods, the spatial error distribution of the fusion positioning method has no obvious bright yellow; the overall color is blue and dark blue, indicating that the proposed method has indeed improved accuracy and stability compared with a single method.

Figure 14 demonstrates the cumulative errors of the five positioning methods based on two group data. As shown in the figure, compared with the multilaterion and the weight least squares positioning methods, the single-point positioning method has better anti-difference performance, but it will still be affected when there is a gross error in the real-time range measurements. The error accumulation curve of the improved matching method is very close to that of the single-point positioning. Moreover, the fusion positioning method combines the advantages of the two methods and is on the far left in the figure. About 53.9% of the positioning error of points are lower than 1 m and 89.6% of the positioning error of points are lower than 2 m based on the group data one and these two indices are 47.2% and 87.6% on group data two.

Table 5 and Table 6 show the detail of the different indices of the five positioning methods based on the two group data set.

As shown in the tables, the statistical results of the proposed methods are better than that of the multilateration positioning and the WLS positioning methods. The mean error of the single-point and improved matching positioning methods are close, which are 1.385 m and 1.344 m, and the corresponding RMSE is 1.936 m and 1.332 m based on the first group data set, respectively. The corresponding indices on group data set two are 1.450 m and 1.461 m, and the corresponding RMSE is 1.742 m and 1.773 m. Among them, the best performance of positioning is the proposed fusion positioning method. The positioning errors of the first 50%, 70%, and 80% of the cumulative error are all smaller than the two positioning methods introduced in the paper. The ME and RMSE of the fusion method are 1.108 m and 1.318 m on the first group data set, respectively. Compared with the single-point and improved matching positioning method, the ME was improved by 20.0% and 21.3%, and the RMSE was improved by 23.4% and 24.8%, respectively. In the second group of data set, the fusion positioning method showed a similar degree of improvement. The ME was improved by 19.1% and 20.0% and the RMSE was improved by 18.7% and 20.8%, respectively. It can be concluded from the experimental results that the proposed method achieved improvement in stability and precision.

## 5. Discussion

The Wi-Fi FTM protocol specified in the IEEE 802.11-2016 standard provides a new two-way ranging approach to estimate the accurate range enhancing the positioning capability. It is necessary to analyze the error characteristics of range measurements. We analyzed the ranging error in different scenarios and found that the ranging error based on the FTM protocol had a systematic error, and the error is related to the clock drift caused by the deviation from the nominal frequency in the equipment. This phenomenon is not consistent with what was previously reported in some papers [30,33], possibly because of the differences in the equipment in the environment used in the experiments.

For the same set of data, it is necessary to use as much information as possible, and the target of this manuscript is consistent with this by applying two positioning methods based on the different principles and integrating them to make full use of the same set of real-time range measurements. In addition to the single point positioning method based on the geometric principle, an improved matching positioning method is also proposed in this paper. According to the characteristics of ranging error, the KNN algorithm was improved to suppress this error. Experimental results show that the positioning effect of this method is equivalent to that of single-point positioning. With the increase in grid spacing, the positioning accuracy will be improved, but the calculation time will also increase. Although the error accumulations of the two positioning methods are very similar, the spatial distribution of the errors is different, making it possible to integrate the two positioning methods. Since the same data are used, the positioning results of the two methods are fused based on the principle of indirect adjustment. Evaluations of the proposed methods and the other two methods show that the fusion positioning method can achieve a better improvement in stability and precision.

The experimental results show that the fusion positioning method is better than the single positioning method. However, this method is based on the accuracy of the two positioning methods. If the indoor environment has a serious impact on multi-path propagation or NLOS, resulting in a large error of the above two positioning methods, the improvement effect of the fusion positioning method will be reduced. The identification of multi-path propagation or NLOS and the studies on the spatial layout of base stations will be of great interest in our future work.

## 6. Conclusions

In this paper, we proposed a fusion positioning method combining a single-point positioning method and an improved matching positioning method based on one real-time range measurement. We first analyzed the error characteristics of range measurements in different scenarios and proposed an abstract ranging model. To make full use of the range measurements, two positioning methods were proposed in this paper. Method one is called the single-point positioning method based on the geometric principle. The other method is an improved matching positioning method similar to the traditional fingerprint positioning method. This paper elaborated on the process of constructing the distance database and an improved matching positioning method is proposed. Although the error accumulations of the two positioning methods are very similar, the spatial distribution of the errors is different, making it possible to integrate the two positioning methods. Since the same data are used, the positioning results of the two methods are fused based on the principle of indirect adjustment. To validate the efficiency of the proposed method, some experiments are carried out at the test site. Experiments show the performance of the improved matching positioning method with a spacing of 1.5 m is close to the single-point positioning method, and the accuracy and stability of the two methods are better than the multilateration and WLS positioning methods. Moreover, the fusion positioning method has the advantages of both methods. Compared with the two methods, the ME, the RMSE, and the other indices of the fusion method are better than that of the single positioning method. It can be concluded from the experimental results that the proposed method achieved improvement in stability and precision. The related issues of identifying and suppressing the NLOS effect in complex indoor environments will be considered in our feature work. The effect on the positioning performance caused by the deployment of the base stations will be studied in our future work.

## Figures and Tables

**Figure 1 sensors-22-03593-f001:**
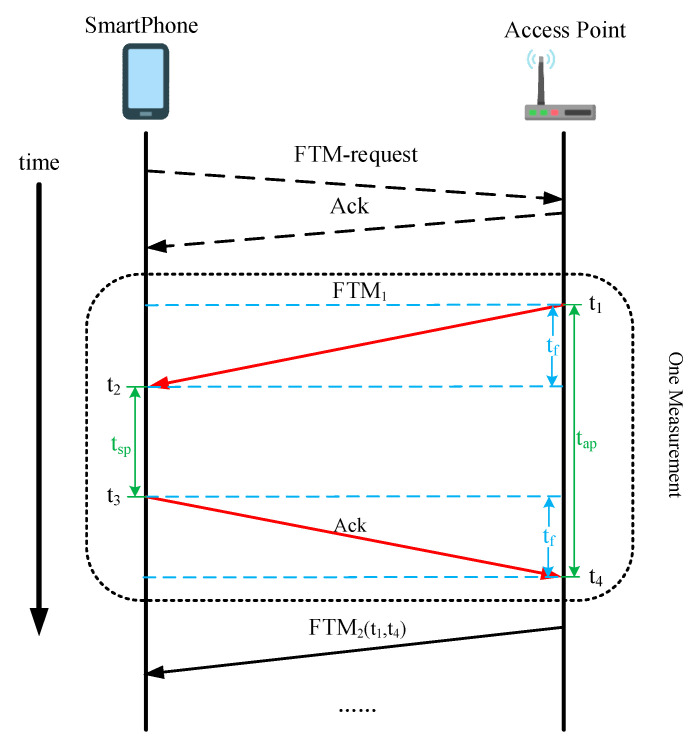
The procedure of the Wi-Fi FTM protocol.

**Figure 2 sensors-22-03593-f002:**
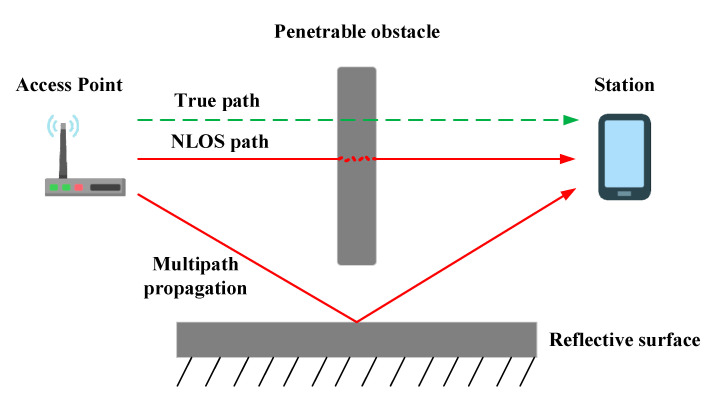
The schematic diagram of the NLOS effect.

**Figure 3 sensors-22-03593-f003:**
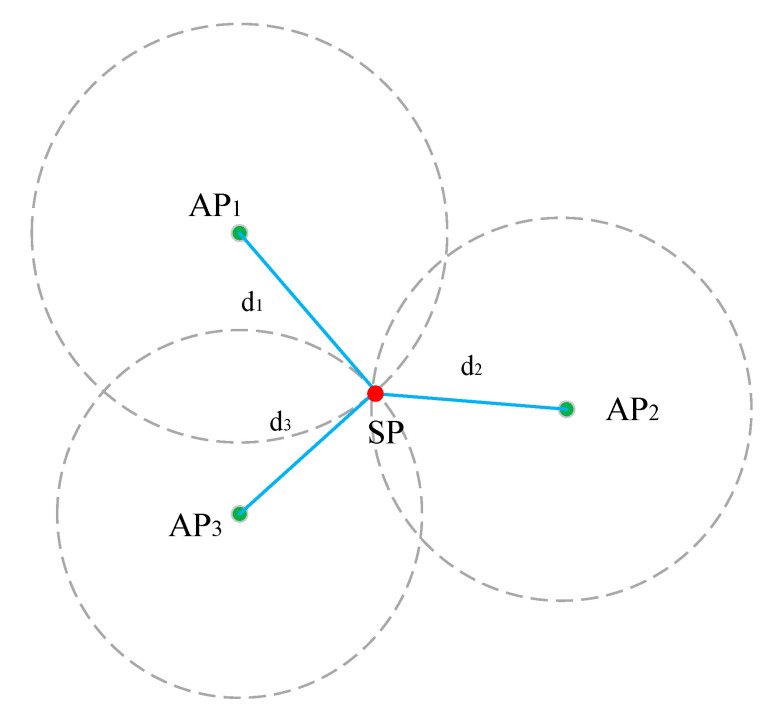
The principle of the geometric positioning method.

**Figure 4 sensors-22-03593-f004:**
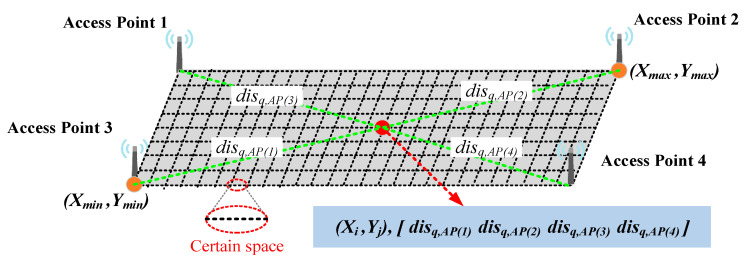
The process of constructing the distance database.

**Figure 5 sensors-22-03593-f005:**
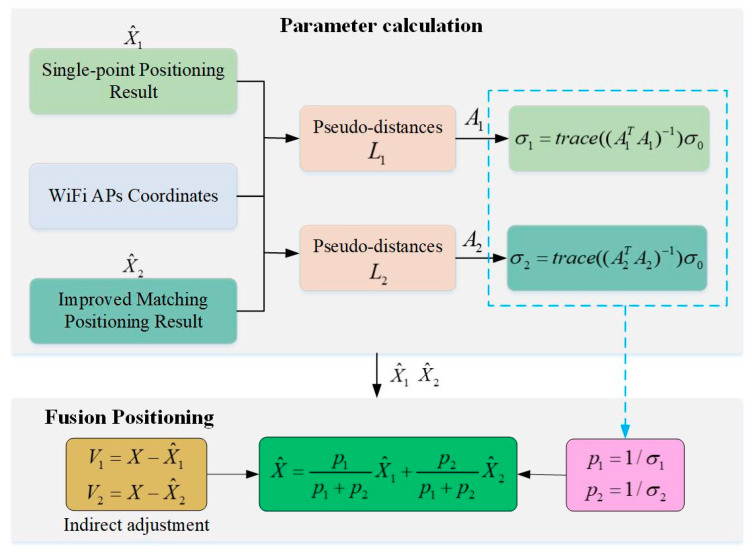
The process of fusion positioning.

**Figure 6 sensors-22-03593-f006:**
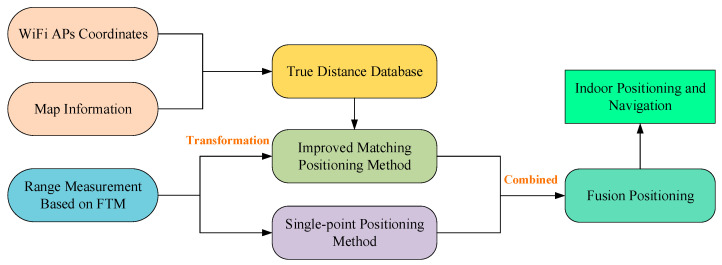
The architecture of the proposed fusion method.

**Figure 7 sensors-22-03593-f007:**
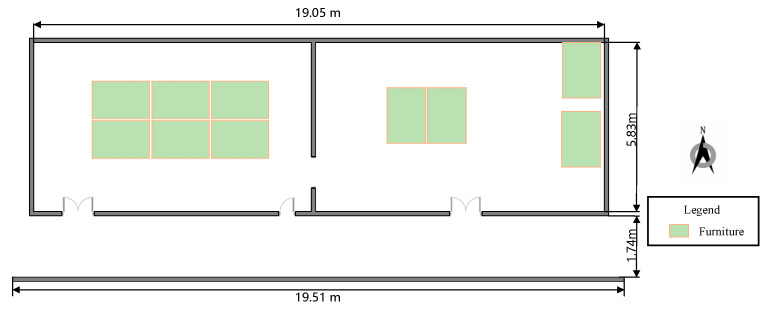
The test site is the 54th Research Institute of China Electronics Technology Group Corporation.

**Figure 8 sensors-22-03593-f008:**
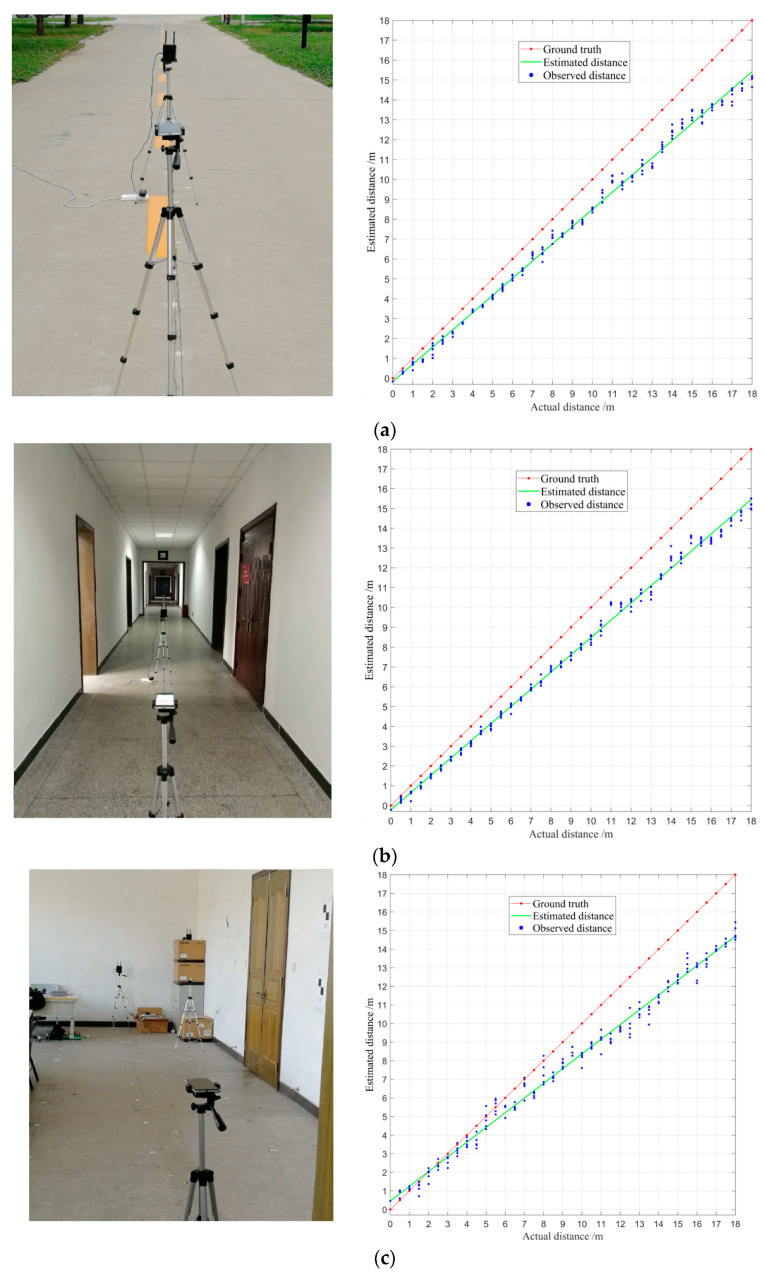
Comparison of real-time range measurements in different scenarios: (**a**) features of real-time range measurements in outdoor setting; (**b**) features of real-time range measurements in the corridor; and (**c**) features of real-time range measurements in the room.

**Figure 9 sensors-22-03593-f009:**
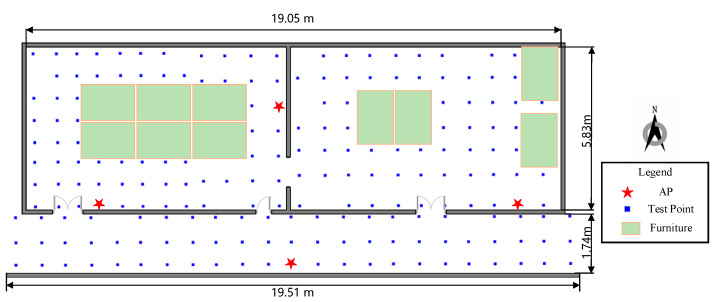
The deployment of APs and the test points in the test site.

**Figure 10 sensors-22-03593-f010:**
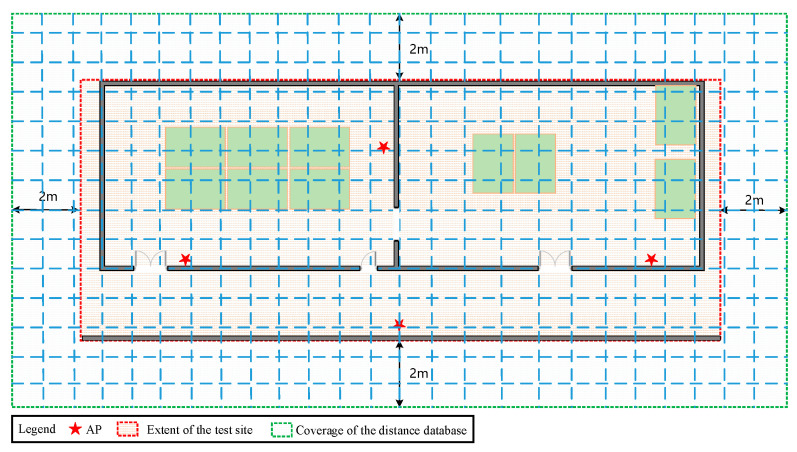
Diagram of the coverage of grid points.

**Figure 11 sensors-22-03593-f011:**
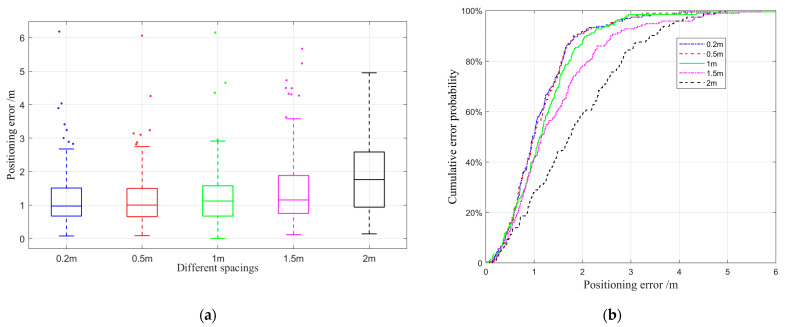
The performance of the improved matching method with different spacing. (**a**) The box plot of the improved matching method with different spacing. (**b**) The cumulative errors of the ranging fingerprint positioning with different spacing.

**Figure 12 sensors-22-03593-f012:**
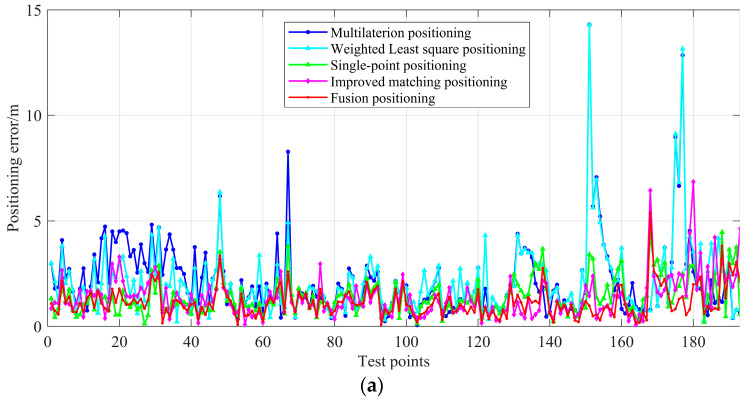
Positioning error of five methods. (**a**) The comparison of positioning error of five methods based on group data set one. (**b**) The comparison of positioning error of five methods based on group data set two.

**Figure 13 sensors-22-03593-f013:**
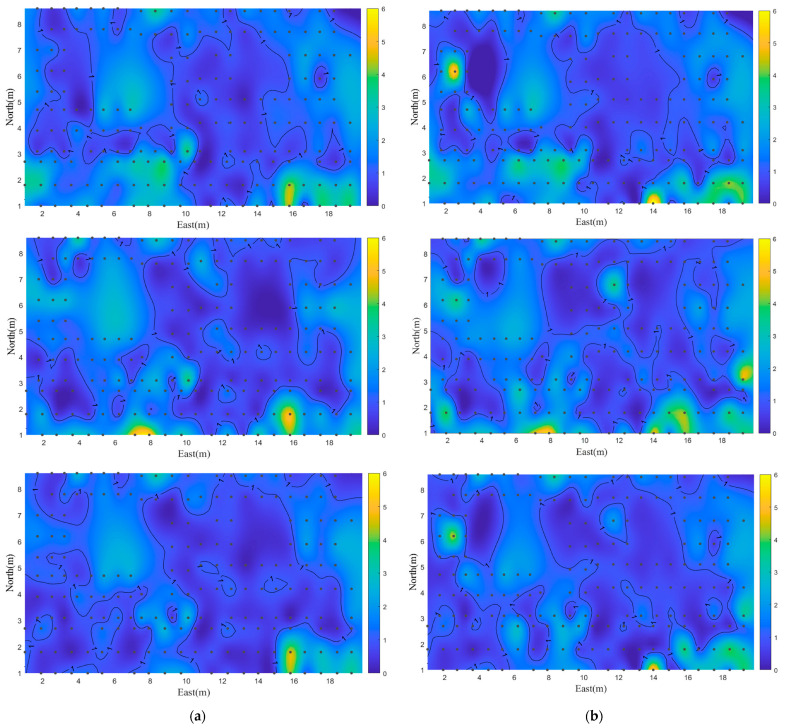
Positioning error spatial distribution of the proposed methods including single-point positioning, improved matching positioning, and the fusion positioning methods from top to bottom. (**a**) Error spatial distribution of t group data set one. (**b**) Error spatial distribution of t group data set two.

**Figure 14 sensors-22-03593-f014:**
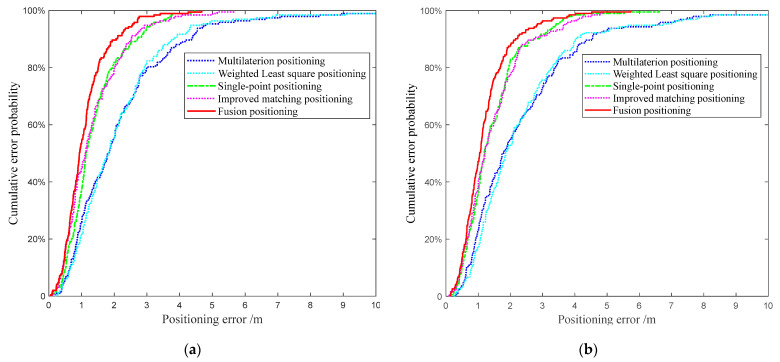
Cumulative errors of different positioning methods. (**a**) The comparison of cumulative errors of five methods based on the first group data set. (**b**) The comparison of cumulative errors of five methods based on the second group data set.

**Table 1 sensors-22-03593-t001:** The pseudocode of single-point positioning algorithm.

Input	Distance measurements set $D=\left[d_{1},\cdots ,d_{n}\right]$ from SP to APs, the coordinates of APs $\left(x_{i},y_{i}\right)$, and initial approximate coordinate $S_{0}=\left(x_{0},y_{0}\right)$.
Process	While for $d_{i}\in D$ construct the approximate distance set $D_{0}=\left[d_{0}^{1},\cdots ,d_{0}^{n}\right]$; construct the observed distance matrix *L*; construct the coefficient matrix A; end for calculate the corrections coordinate ∆; according to Equation (14); update the approximate coordinate $S_{0}$ based on Equation (15); until meeting the condition of stop end while
Output	S, the final coordinate of the SP

**Table 2 sensors-22-03593-t002:** The distance database.

Id	Coordinate	Distance Fingerprint Item
1	$\left(x_{0},y_{0}\right)$	$\left[dis_{1,AP\left(1\right)},dis_{1,AP\left(2\right)},\ldots ,dis_{1,AP\left(k\right)}\right]$
2	$\left(x_{0},y_{1}\right)$	$\left[dis_{2,AP\left(1\right)},dis_{2,AP\left(2\right)},\ldots ,dis_{2,AP\left(k\right)}\right]$
…	…	…
m×n	$\left(x_{m},y_{n}\right)$	$\left[dis_{m\times n,AP\left(1\right)},dis_{m\times n,AP\left(2\right)},\ldots ,dis_{m\times n,AP\left(k\right)}\right]$

**Table 3 sensors-22-03593-t003:** The entry number of the distance database and running time comparison.

Spacing	Number	Mean Running Time/(s)
0.2 m	6844	0.635
0.5 m	1128	0.125
1.0 m	288	0.031
1.5 m	128	0.014
2.0 m	72	0.007

**Table 4 sensors-22-03593-t004:** Positioning error comparison/(m).

Spacing	50%	70%	80%	ME	RMSE
0.2 m	0.969	1.361	1.389	1.144	1.381
0.5 m	0.997	1.379	1.557	1.146	1.369
1.0 m	1.121	1.502	1.721	1.237	1.475
1.5 m	1.151	1.706	2.089	1.444	1.765
2.0 m	1.759	2.412	2.804	1.861	2.161

**Table 5 sensors-22-03593-t005:** Positioning error comparison based on the first group data set/(m).

Method	50%	70%	80%	ME	RMSE
Multilateration	1.795	2.589	2.991	2.216	2.907
Weighted least square	1.822	2.601	2.918	2.168	2.809
Single-point	1.142	1.581	1.909	1.385	1.626
Improved matching	1.118	1.628	2.000	1.344	1.645
Fusion	0.934	1.242	1.503	1.108	1.318

**Table 6 sensors-22-03593-t006:** Positioning error comparison based on the second group data set/(m).

Method	50%	70%	80%	ME	RMSE
Multilateration	1.789	2.798	3.356	2.401	3.198
Weighted least square	1.848	2.729	3.244	2.416	2.192
Single-point	1.184	1.715	1.941	1.450	1.742
Improved matching	1.208	1.746	2.102	1.461	1.773
Fusion	1.045	1.361	1.666	1.217	1.468

## Data Availability

The experiment uses an internal data set and the data presented in this study are available on request from the corresponding author.

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
