# Peer review of "A Two-Step Fusion Method of Wi-Fi FTM for Indoor Positioning"

_sensors, 2022, doi:10.3390/s22093593_

Round 1

Reviewer 1 Report

The manuscript “ A Two-Step Fusion Method of WiFi FTM for Indoor Positioning” addresses a topic of interest to a broad audience and fits the scope of this special issue of the journal.

This article designs two positioning steps and proposed a fusion method to refine the performance of indoor positioning. From the experimental results that the proposed method achieved improvement in stability and precision.

  1. Page 10 “Table 1 Process …according to Equation(17)…. according to Equation(18)”; Are they right?
  2. Page 12 line 353 “… with the expected value in Equation (22)”; It should be Equation (20).
  3. Page 16 line 452 It looks like in the room not in the outdoor.(Figure 9. (c))
  4. Page 19 line 543 It would be (b) not (a).

Author Response

Response to the Review 1 Comments

Dear Editors and Reviewer:

We especially appreciate your careful reading, helpful comments, and constructive suggestions, which has significantly improved the presentation of our manuscript. We have carefully considered all comments from the reviewers and revised our manuscript thoroughly. The manuscript has also been double-checked, and the typos and grammar errors we found have been corrected. Based on these comments and suggestions, we have made careful modifications on the original manuscript. We hope the new manuscript will meet your magazine’s requirement. You will find our point-by-point responses to the reviewer’ comments/questions in the following. Thank you again for your kind help.

Comments from Reviewer 1

1. Page 10 “Table 1 Process …according to Equation(17)…. according to Equation(18)”; Are they right?

Thanks for the constructive comments. We have modified the corresponding Equations number (17), (18) in the sentences in the pseudocode of single-point positioning algorithm of Table 1 as (14) and (15) respectively.

2. Page 12 line 353 “… with the expected value in Equation (22)”; It should be Equation (20).

Thanks for the comments. We have changed the Equation (22) to Equation (20) according your correction.

3. Page 16 line 452 It looks like in the room not in the outdoor.(Figure 9. (c)).

Thank for your suggestions. The Figure 9 (c) refers to the experimental results of the real-time range measurements in the room and we have changed the expression in page 15 line 554.

4. Page 19 line 543 It would be (b) not (a).

Thanks for the comments. We have made careful modifications according to your comments in page 18 line 662.

Reviewer 2 Report

Dear Authors,

  I have read your manuscript with great interest as it describes an extensive study of the accuracy of positioning algorithms for location measurements obtained with WiFi FTM technology. The methodology, implementation of the experiments and the results are fairly well presented and described. At certain points, however, a particular theoretical description and state of art could be presented in a more concise manner. More detailed:

  1. The Introduction section could be slightly reduced. Especially the contents presented in line (~40 - ~80) can be summarized, as they are presented also in section 2.
  2. I propose to avoid to the figures in introduction section. The fig 1 and related text can be a part of section 3.
  3. The content in section 3.1 presents the basic theory rather and can be omitted or presented in more compressed form. In particular, paragraph starting in line 237 starts to present some results – these are not adequate to the section Materials and methods. Moreover, the results seems to be duplicated in subsection 4.2 (figure 9).
  4. In 3.2. subsection the single-point positioning method is described in detail, but it’s looks like trilateral (multilateral) positioning. So, if described method has no essential modifications, it’ worth to consider of reduction this section. In similar way, the next subsection presents some kind of fingerprints method, where kNN algorithm was proposed as classifier.
  5. Please explain the difference between formula (9) and (19), why the dem error was skipped ?
  6. What is the difference between single point and multilateral methods used in subsection 4.4? From sentence (line 289-90) it can be concluded that it could be the same method.

Minor:

  1. Line 193 – explain the acronym AOA (Angle of Arrival ??)
  2. Some numerations are duplicated (Table 3, subsection 3.2)
  3. Figure 9c caption – should be “room” rather than “outdoor”
  4. Improve some figures quality (axis caption, legend) – Figure 14, 15

Author Response

Response to the Review 2 Comments

Dear Editors and Reviewer:

We especially appreciate your careful reading, helpful comments, and constructive suggestions, which has significantly improved the presentation of our manuscript. We have carefully considered all comments from the reviewers and revised our manuscript thoroughly. The manuscript has also been double-checked, and the typos and grammar errors we found have been corrected. Based on these comments and suggestions, we have made careful modifications on the original manuscript. We hope the new manuscript will meet your magazine’s requirement. You will find our point-by-point responses to the reviewer’ comments/questions in the following. Thank you again for your kind help.

Comments from Reviewer 2

1. The Introduction section could be slightly reduced. Especially the contents presented in line (~40 - ~80) can be summarized, as they are presented also in section 2.

Thanks for your constructive comments. We have thoroughly revised the introduction section. We have deleted the parts that are similar and repetitive to the second 2, and simplified the corresponding expressions in section 1.

2. I propose to avoid to the figures in introduction section. The fig 1 and related text can be a part of section 3.

Thanks for your valuable suggestions. We have moved Figure 1 from the introduction section and the corresponding text to the section 3 of the manuscript and modified the description of the the architecture of the two-step fusion positionig in the lines 502-508.

3. The content in section 3.1 presents the basic theory rather and can be omitted or presented in more compressed form. In particular, paragraph starting in line 237 starts to present some results – these are not adequate to the section Materials and methods. Moreover, the results seems to be duplicated in subsection 4.2 (figure 9).

Thanks for your constructive comments. We have simplified the basic theory of the WiFi FTM protocol in subsection 3.1. Moreover, the content starting in lin 237 is repeated in the section 4. We deleted Figure 2 and referred to Figure 8 (a) in line 259. The performance of the error  have been redescribed according to the Figure 8 (a).

4. In 3.2. subsection the single-point positioning method is described in detail, but it’s looks like trilateral (multilateral) positioning. So, if described method has no essential modifications, it’ worth to consider of reduction this section. In similar way, the next subsection presents some kind of fingerprints method, where kNN algorithm was proposed as classifier.

Thanks for your constructive comments. Although the multilateral positioning and single-point positioning are geometric positioning methods, there are differences between the two approaches. The mutilateral positioning method resolves the problem from the pespective of the squaring and linearization, and only needs to calculate once to get the positioning result. The single-point positioning method directly starts from the nonlinear equation and linearizes the equation with Taylor's expansion to solve the problem, but it needs to iterate several times to obtain the positioning result. Compared with the multilateration positioning, the single-point positioning can resist coarse ranging errors and has the characteristic of numerical stability. In addition, we have simplified the contents of KNN classifier alogrithm in lines 387-389.

5. Please explain the difference between formula (9) and (19), why the dem error was skipped ?

Thanks for your constructive comments. In fact, the multipath effect is seriously affected by the environment, so it is difficult to find a clear way to describe the error.  The improved matching method proposed in this paper is aimed at suppressing error B. Therefore, we supplement the hypothesis in this manuscript in lines 391-393, assuming that the observed distance does not contain multipath error in a good electromagnetic environment. We also changed equation 18, changing the equal sign to approximately equal.

6. What is the difference between single point and multilateral methods used in subsection 4.4? From sentence (line 289-90) it can be concluded that it could be the same method.

Thanks for your constructive comments. There are differences between the two approaches. The two methods solve problems in different ways, and the specific differences can be found in Equations 10,12 and 13. The mutilateral positioning method resolves the problem from the pespective of the squaring and linearization, and only needs to calculate once to get the positioning result. The single-point positioning method directly starts from the nonlinear equation and linearizes the equation with Taylor's expansion to solve the problem, but it needs to iterate several times to obtain the positioning result. These two methods have differences in computational efficiency and accuracy. Compared with the multilateration positioning, the single-point positioning can resist coarse ranging errors and has the characteristic of numerical stability.

7. Line 193 – explain the acronym AOA (Angle of Arrival ??).

Thanks for your constructive comments. The acronym AOA is the term of angle of arrival. We have modified the content in line 207.

8. Some numerations are duplicated (Table 3, subsection 3.2).

Thanks for your constructive comments. We have carefully revised the subsection 3.2 and modified the duplication of some numerations. We also changed the duplicate table 3 to Table 4

9. Figure 9c caption – should be “room” rather than “outdoor”.

Thanks for your constructive comments. he Figure 9 (c) refers to the experimental results of the real-time range measurements in the room and we have changed the expression in page 15 line 554.

10. Improve some figures quality (axis caption, legend) – Figure 14, 15.

Thanks for your constructive comments. We redrew Figure 14 and 15 to improve the quality of the graphics and make the axes caption, legends and so on look clearer.

Reviewer 3 Report

This paper studies the WiFi location method in indoor location. Based on the full use of WiFi FTM measurement, an improved matching algorithm and fusion location method are proposed to improve the stability and accuracy of indoor location. This paper is complete in structure and innovative, but it is not perfect in some detail places. My detailed comments are as follows.

  1. The relationship between the estimated distance and the observed distance is not clearly stated in Figure 3(b)
  2. Equation (9) does not clearly state the meaning of
  3. Line 350 states that "When the range measurement d Ì‚ is considered to be composed of the true distance and the error ". But Equation 19 contains
  4. line 496 'The value ? of the KNN algorithm is chosen as 5', the meaning of the sentence is unclear
  5. Figure 14 (b) font size is not uniform
  6. Line 443 figure number expression is not clear
  7. The control experiment of the two sets of data sets is not sufficient
  8. This paper fails to fully show the different positioning accuracy and error distribution between single point positioning and improved matching algorithm

Author Response

Response to the Review 3 Comments

Dear Editors and Reviewer:

We especially appreciate your careful reading, helpful comments, and constructive suggestions, which has significantly improved the presentation of our manuscript. We have carefully considered all comments from the reviewers and revised our manuscript thoroughly. The manuscript has also been double-checked, and the typos and grammar errors we found have been corrected. Based on these comments and suggestions, we have made careful modifications on the original manuscript. We hope the new manuscript will meet your magazine’s requirement. You will find our point-by-point responses to the reviewer’ comments/questions in the following. Thank you again for your kind help.

Comments from Reviewer 3

1. The relationship between the estimated distance and the observed distance is not clearly stated in Figure 3(b)

Thanks for the constructive comments. We deleted Figure 2 and referred to Figure 8 (a) in line 259. We have added the content in lines 261-265 to clearly describe the relationship between the estimated distance and the obsereved distance.

2. Equation (9) does not clearly state the meaning of.

Thanks for the comments. We have added some expressions to clearly described the meaning of three types of errors.

3. Line 350 states that "When the range measurement d Ì‚ is considered to be composed of the true distance and the error ". But Equation 19 contains.

Thanks for your constructive comments. In fact, the multipath effect is seriously affected by the environment, so it is difficult to find a clear way to describe the error.  The improved matching method proposed in this paper is aimed at suppressing error B. Therefore, we supplement the hypothesis in this manuscript in lines 391-393, assuming that the observed distance does not contain multipath error in a good electromagnetic environment. We also changed equation 18, changing the equal sign to approximately equal.

4. line 496 'The value ? of the KNN algorithm is chosen as 5', the meaning of the sentence is unclear.

Thanks for your constructive comments. We changed the expression to make the meaning clearer. The corresponding content is expressed as follow: The value of K of the KNN algorithm is chosen as 5.

5. Figure 14 (b) font size is not uniform.

Thanks for your constructive comments. We redrew Figure 14 quality of the graphics and unified the fonts.

6. Line 443 figure number expression is not clear.

Thanks for your constructive comments. We have modified the correct figure number in the manuscript.

7. The control experiment of the two sets of data sets is not sufficient.

Thanks for your constructive comments. We collected several groups of data, and the performance of each group of data is similar. However, putting all the experimental results in the paper would be a bit redundant. To make the paper more concise and easy to understand, we randomly selected two groups of experimental results to demonstraate. 

8. This paper fails to fully show the different positioning accuracy and error distribution between single point positioning and improved matching algorithm.

Thanks for your constructive comments. The error distribution between single point positioning and improved matching algorithm had been demonstrated by contour lines with the position error of 1m in the Figure 13. As can be seen from the figure, the difference between the left and the upper right corner of the figure is particularly obvious of two positioning methods. We will find a better way to comprehensively describe the spatial error distribution of different methods in the future research.
